# Automated Monitoring of Panting for Feedlot Cattle: Sensor System Accuracy and Individual Variability

**DOI:** 10.3390/ani10091518

**Published:** 2020-08-27

**Authors:** Md Ashraful Islam, Sabrina Lomax, Amanda K. Doughty, Mohammed R. Islam, Cameron E. F. Clark

**Affiliations:** 1School of Life and Environmental Sciences, University of Sydney, Camden 2570, Australia; sabrina.lomax@sydney.edu.au (S.L.); md.islam@sydney.edu.au (M.R.I.); cameron.clark@sydney.edu.au (C.E.F.C.); 2Department of Dairy Science, Faculty of Animal Science and Veterinary Medicine, Patuakhali Science and Technology University, Dumki, Patuakhali 8602, Bangladesh; 3Allflex Australia Pty Ltd., 33 Neumann Road, Capalaba 4157, Australia; Amanda.Doughty@allflex.com.au

**Keywords:** ear tag sensor, accelerometer, heat stress, validation, individual variation

## Abstract

**Simple Summary:**

Panting is considered a robust animal response indicator of heat stress; however, continuous visual monitoring is impractical on a commercial scale. Current thermal indices are good predictors of heat stress at a herd level but have limited application at the individual level. The automated monitoring of heat stress responses of individual cattle based on animal response and environmental parameters are required for strategic management and genetic selection. We validated an ear tag-based sensor for the monitoring of panting in cattle and determined the individual variability in panting duration for heat stress events, comparing with existing thermal indices. Sensors were able to monitor differences in cattle panting due to breed, coat colour, and individual variability. High variation in panting responses between and within genotype and coat category highlights the opportunity for targeted heat amelioration based on breed and coat colour grouping of animals, and selection within breeds and types for heat resilience.

**Abstract:**

Heat stress causes significant economic losses by reducing the productivity and welfare of cattle whilst requiring a significant investment in resource for amelioration. Panting score (PS) is considered a robust indicator of cattle heat stress; however, individualised visual monitoring is impractical. Thermal index-based monitoring and mitigation decisions are applied at the herd level, but they have limited application for the individual animal. As such, an automated system to monitor the real-time animal response to heat stress is required for strategic mitigation. Our objectives were to validate an accelerometer-based ear tag sensor to monitor cattle panting and to determine individual variability in heat stress responses with reference to thermal indices. Two experiments were conducted: Experiment 1 validated the sensors, and Experiment 2 determined individual variability comparing sensor data against thermal indices. Ear tag sensors were fitted at feedlot entry to continuously monitor the behaviour of 100 steers of mixed breed in Experiment 1 and 200 steers and heifers of mixed breed in Experiment 2. Sensor-derived ‘heavy breathing’ was validated against visually observed PS. Sensor-derived behaviour bouts were analysed as ‘raw’, and single behaviour states were also converted to the preceding bout of ≥2 min, which was referred to as ‘fill’ data for the validation study. Our results demonstrate the sensors’ ability to accurately monitor panting in feedlot cattle. Sensor-recorded ‘heavy breathing’ duration per animal was highly correlated to observed panting duration for both raw (r = 0.89) and fill (r = 0.90) data; however, the concordance correlation co-efficient was lower for raw (0.45) as compared with fill (0.76). Predicted agreement for raw data were 75%, 45%, and 68% and predicted agreement for fill data were 65%, 54%, and 83% for PS0, PS1, and PS2, respectively. Sensitivity for raw data were 39%, 37%, and 45% and for fill data, they were 59%, 54% and 82% for all PS data, PS1 and PS2, respectively. Specificity and positive predictive values for both raw (77% and 79%, respectively) and fill (65% and 77%, respectively) data show the probability of reporting false positives by sensors to be low. Experiment 2 revealed that the duration of panting increased from 0800 to 1700 h alongside changes in thermal indices with significant differences between and within breed and coat colour categories of cattle, suggesting that grouping and allocating heat amelioration measures by breed and coat colour can be effective in commercial feedlots. However, there was high variability (CV > 80%) in the duration of panting between individuals within the same breed and same coat colour, revealing the potential for strategic management at an individual level, and with the same data, genetic selection for heat resilience.

## 1. Introduction

Heat stress is “the sum of external forces acting on an animal that causes an increase in body temperature and evokes a physiological response” [1] or the “imbalance between metabolic heat production inside the animal body and its dissipation to the surroundings under high air temperature and humid climates” [2]. The degree of heat stress depends on animal factors including genotype, coat colour, coat type, sex, body condition, health, physiological and metabolic condition; and environmental factors including temperature, humidity, solar radiation, wind speed, cloud cover, rainfall, and management [3]. Heat stress is associated with significant economic loss [4,5,6,7,8] in the cattle industry, compromised production [5,9,10,11] and welfare [12,13] and in extreme cases, a greater incidence of mortality [4,14]. In addition, ameliorating heat can require a significant amount of infrastructure, labour, and energy [7].

Heat stress has been traditionally monitored using the visual assessment of an animal’s response evoked by high ambient temperatures and 25 °C has been suggested as a threshold after which both dairy [15] and beef [16] cattle experience heat stress. However, a threshold of 22 °C [17] has also been suggested for feedlot cattle. Cattle, similar to other homeothermic animals, attempt to maintain a constant body temperature of approximately 38–39 °C [18,19,20]) by dissipating heat produced in the body into the surrounding environment through conduction, convection, evaporation, and radiation [21,22]. Non-evaporative heat dissipation mechanisms become less effective, and animals depend more on evaporative heat dissipation, when ambient temperatures rise to 29–32 °C and approach, or exceed, skin surface temperature (approximately 37 °C) [23,24]. However, evaporative heat dissipation can be increased significantly at temperature as low as 18.3 °C [22]. Evaporative heat dissipation is achieved in part by sweating and further by increased respiration rate with characteristic shifts in respiratory dynamics (measured as panting scores [17,18,25,26]), whereby body heat is released as latent heat of vapourisation of moisture from the skin surface and respiratory tract mucosa [24,27,28]. As such, monitoring respiration rate has become a reliable animal response indicator of heat stress [4,29]. However, visual recording of respiration rate is impractical under commercial and field conditions [30]. The inadequacy of ambient temperature as a single measure to represent the thermal condition of cattle has led to the development of numerous thermal indices such as the temperature humidity index, THI; black globe humidity index, BGHI; environmental stress index, ESI; skin surface temperature humidity index, STHI; respiratory rate predictor, PRR; heat load index, HLI; and the accumulated heat load unit, AHLU as measures of the impact of heat on the animal [25,26,31,32].

The temperature humidity index, combining both air temperature and relative humidity, was developed based on Thom’s discomfort index for humans [33] and has been widely used in livestock since the 1960s with its limitations related to wind speed and solar radiation [26,34]. Latter indices such as ESI combining air temperature, relative humidity, and solar radiation and HLI combining relative humidity, wind speed, and black globe temperature accounted for both environmental stressors and physiologic response, which were modified for animal and management factors. These latter indices are better correlated with the respiration rate and rectal temperature of cattle as compared with THI [32]. The HLI is currently used by the Australian feedlot industry and was developed with reference to panting score, which is a robust indicator of heat stress at a herd or pen level [17,35,36]. Panting scores range between 0 and 4.5 and are assigned on the basis of severity of panting, presence or absence of drool, open mouth, and extended tongue [25]. The robustness of HLI is due to the inclusion of several environmental (black globe temperature accounting for ambient temperature and solar radiation, humidity, and wind speed), management (cooling effect, manure pad status, days on feed and drinking water temperature) and animal factors (genotype, coat colour, and health status) associated with heat load [25]. Despite its robustness, HLI does not represent the accumulated effects of heat exposure and the heat stress level of individual cattle. The adjustment of HLI thresholds based on animal factors and the combination of intensity and duration of heat exposure in the form of AHLU is now practiced for a better assessment of thermal condition at the group level. However, microclimatic variations between locations of the pen, compared to the location of the weather station, may affect the assessment of thermal condition at the individual level. To determine the impact of heat load on the individual animal across time, and for the strategic mitigation of cattle heat stress at an individual level, automated monitoring of the real-time animal response to heat stress is required. Commercially available accelerometer-based sensors (HR-LDn; SCR Engineers Ltd., Netanya, Israel) monitor the behaviour of cattle [37,38], including ‘heavy breathing’ (referred to as ‘panting’ in the latter sections), as defined by forward–backward heaving with an increased rate of respiration and thoracic wall extension [39]. Accelerometer-based ear tag sensors have been previously validated to monitor cattle behaviour and provide a promising tool for the automated monitoring of eating, ruminating, and resting behaviours [40,41,42]. However, ear tag sensors require validation to monitor and quantify heat stress-related panting behaviour and individual variability.

The degree of heat stress varies with genotype and coat colour, with *Bos taurus*, dark coloured (black and red) and high coat scored cattle less heat tolerant than *Bos indicus*, light coloured and low coat score cattle [21,23,43,44]. As oppose to the common belief that *Bos indicus* cattle do not need heat load alleviation measures, Brahman cattle have been found to increase shade utilisation by 27% between 0800 and 1200 h [45]; however, the variability between cattle within genotype or coat categories is less known. Continuous and long-term monitoring is required to quantify individual variability in heat stress responses for more specific mitigation decisions and for improved genetic selection for heat resilience. Our objectives were to (i) validate an accelerometer-based ear tag sensor for cattle panting, (ii) determine the variability in panting duration between cattle breeds and individuals, and (iii) compare thermal indices against sensor-derived panting at an individual animal level.

## 2. Materials and Methods

These experiments were conducted at a commercial feedlot in Southeastern Queensland, Australia between October 2018 and February 2019. All experimental protocols were approved by The University of Sydney Animal Ethics Committee (No. 2017/1213). Experiment 1 validated an ear tag-based sensor for panting. Experiment 2 determined the variability in panting through time between breeds and individuals, and how sensor-derived panting behaviour varied with thermal indices.

### 2.1. Experiment 1

#### 2.1.1. Animals and Feedlot Entry

A pen (44 × 22 m^2^) of 100 mixed breed steers (Brangus, Santa Gertrudis, Droughtmaster, and Charbray) with a mean liveweight of 378 ± 2.87 kg were used. On entry to feedlot (day 1), cattle were fitted with visual identifiers (ear tag ID and a large number painted on the side of each animal) and with ear tag sensors (HR-LDn Allflex eSense™, Heatime^®^ Pro+ software, SCR Engineers Ltd., Netanya, Israel) to monitor visual and sensor-derived animal behaviour, respectively. The sensors were installed with an ear tag applicator in the ear that was not attached with an NLIS (national identification system) tag; the sensor ID was matched with the Cattle ID tag, and the whole sensor unit was disinfected and then fitted in the centre of the left ear as per the manufacturer’s guideline (https://www.youtube.com/watch?v=p75RUMk_nuI). As per standard protocol for this feedlot, a visual ID tag (1 piece, self-piercing) was fitted in the inner part (from centre) of the same ear (Figure 1).

#### 2.1.2. Feed and Feeding Schedule

Cattle were offered an ad libitum feedlot transition ration (Ration 1, Table 1) at 0800 h for day 1 and day 2 and then from day 3 onwards were offered Ration 1 at 0900 h and Ration 2 at 1300 h (Table 1).

#### 2.1.3. Climate Data

Climate data were recorded at 10 min intervals from an on-site weather station to calculate THI following NRC recommendations, THI = (1.8 × dry bulb temperature + 32) − ([0.55 − 0.0055 × relative humidity] × (1.8 × dry bulb temperature − 26)) [46] and HLI was calculated as per Gaughan et al. [25], HLI_BG>25_ = 8.62 + (0.38 × RH) + (1.55 × BG) − (0.5 × WS) + e^(2.4−WS)^, and HLI_BG<25_ = 10.66 + (0.28 × RH) + (1.3 × BG) − WS. Average daily maximum temperature (°C) and relative humidity (%) for Experiment 1 was 33 and 81 (Table 2), respectively. Average THI and HLI values across 24 h shows that the hotter part of the day was between 0800 and 1900 h (Figure 2).

#### 2.1.4. Sensor-Based Monitoring

Individual cattle sensor-derived data were recorded and sent to a base station to enable the categorisation of animal response in minute intervals according to a proprietary algorithm (optimised for mature dairy cattle) for both experiments. The animal response recorded in each minute was one of the following: panting or “other” (eating, rumination, resting, and activity level). Minutes with no specific behaviour pattern were classified as undefined.

#### 2.1.5. Visual Observation

Visual observations were continuously conducted between 0800 and 1700 h with a maximum observation duration of 15 min/animal on days 5 to 8 post feedlot entry focusing on panting as defined by forward–backward heaving with an increased respiration rate and extension of the thoracic wall in cattle. Panting scores were assigned as per Table 3. Two observers recorded behaviour in Experiment 1. Observers were trained to use the observation software and the behaviour ethograms (Table 3) before starting actual data collection. Inter-observer reliability on assigning panting scores was calculated as Cohen’s kappa [47] (κ = 0.91) from 188-min independent observations by two observers on the same subjects for the same durations. One observer was located 10 m from the feed bunk at an elevated position to observe the entire pen. The second observer was located at the rear of the pen near the water trough to ensure that all cattle could be viewed. Observers recorded individual cattle behaviour directly into a tablet device using customised software (SCR Engineers Ltd., Netanya, Israel). Timestamps on the tablet device were synchronised daily with the sensor software before recording commenced. Each observer opportunistically selected an individual animal according to panting categories as described in Table 3; priority was given to find animals with panting scores 1 or greater due to the fact that moderate heat stress conditions evokes panting response to only a certain percentage of cattle. When no cattle were identified as panting, observation was performed on selected non-panting cattle. Selection was done for animals that were clearly visible to the observer. Animal ID was entered into the observation software, and a base behaviour was selected to start the recording. An individual animal was observed continuously for up to 15 min, with behaviour recorded as panting or other. When an animal moved out of sight, with the head and/or body not visible to the observer, the observation was ended with the last minute recorded as undefined, and a new animal then selected. 

Two observers recorded 86 panting events (22 for PS 0, 44 for PS 1, and 20 for PS 2) comprising 1290 min of observations on 42 cattle. Each observer independently recorded observations. To avoid duplication of observations between observers, different animals were selected for each observer and repetition of the same animal for the same day was avoided.

#### 2.1.6. Data Organisation and Manipulation

Observed and raw sensor data were collected between 18 and 21 February 2019. Sensor-derived behaviour state data were organised at a minute level and aligned with observation data. Undefined behaviour states (both sensor and observer recorded) were excluded (*n* = 303 total, tag = 13, observers = 290) from the dataset, and the remaining 987 (657 panting and 330 non-panting) minutes of behaviour state data were used for the analysis. Any behaviour state other than panting was considered as non-panting.

Alongside the use of ‘raw’ minute level data, the same sensor data were manipulated such that any single minute state after 2 consecutive records of the same state was filled with the preceding state until the next 2 consecutive records of a differing state were recorded. This was termed ‘fill’ behaviour states. This ‘fill’ approach was evaluated as an animal may perform several behaviours in a single minute, including heavy breathing, and still not necessarily be recorded as heavy breathing during that minute.

### 2.2. Experiment 2

#### 2.2.1. Animals and Feedlot Entry

A pen (44 × 44 m^2^) of 200 mixed breed *Bos indicus* × *Bos taurus* cattle (categorised into 2 groups: 50% *B. indicus* (BI) and less than 40% BI, based on level of BI as described in the animal fact sheet provided by vendors), mixed sex and coat colour (dark consisting black and/or red, tan and white) with a mean liveweight of 342 ± 1.84 kg were used. There were 108 cattle categorised as less than 40% BI, and of these, 27 were steers (15 dark (red) and 12 tanned colour) and 81 were heifers (66 dark (58 red and 8 black), 10 tanned and 5 white). There were 92 cattle categorised as 50% BI, and of these, 14 were steers (6 dark (black), 3 tanned and 5 white) and 78 were heifers (18 dark (15 black and 3 red), 23 tanned and 37 white). These cattle received similar treatments on entry to the feedlot as per Experiment 1. Cattle docility scores [17,48] were assessed in 1 to 5 categories based on cattle temperament and behaviour when entering and exiting the crush with a score of 1 being assigned to cattle that were calm and quiet in the crush and on exit, and 5 for cattle that were highly agitated and nervous.

#### 2.2.2. Feed and Feeding Schedule

Cattle were adapted to the final finishing diet through the provision of three transitional diets (Table 4). Cattle were offered (ad libitum) Ration 1 on day 1, Ration 2 on days 2 to 5 and Ration 3 on days 6 to 10 at 0800 h. Ration 3 was offered as 60%, and finisher ration 4 was offered as 40%, of daily allowance at 0800 and 1300 h, respectively, on day 11 to 14. From day 15, ration 4 was offered at 0800 h, 1000 h, and 1300 h respectively at 20%, 20% and 60% of the daily feed allowance.

#### 2.2.3. Climate Data

Climate data were collected and analysed as per the procedure described for Experiment 1. Daily temperature and relative humidity, and hourly THI and HLI values are provided in Table 5 and Figure 3, respectively.

#### 2.2.4. Sensor-Based Monitoring

Individual cattle sensor-derived data were recorded and sent to a base station to enable the categorisation of behaviours in minute intervals as described for Experiment 1.

#### 2.2.5. Visual Observation

Visual observations of cattle behaviour and activity were performed at 0800 h, 1200 h, and 1600 h for general monitoring purposes throughout the experimental period.

#### 2.2.6. Data Organisation and Manipulation

Raw sensor data were collected between October 2018 and January 2019. Heat event periods were selected with the requirement of at least 3 consecutive days with maximum THI and HLI values equal to or greater than 90 (Figure 4) for early (days 17 to 21 on feed, 3 to 7 November), mid (days 43 to 47 on feed, 29 November to 3 December) and end (days 63 to 66 on feed, 19 to 22 December) feed periods. Sensor data of the selected three heat event periods were analysed.

### 2.3. Statistical Analyses

All analyses (Experiment 1 and 2) were performed using the statistical package of Genstat (GenStat for Windows, 18th Edition, VSN International 2015, Hemel Hempstead, UK). Statistical significance was determined at *p* < 0.05.

#### 2.3.1. Experiment 1

To determine the level of agreement between sensor-derived (raw and fill) and observed panting, a generalised linear mixed model (GLMM) was used with binomial distribution and a logit link function. Panting score of the individual animal was included as a fixed effect and animal as a random term in the model. The model was as follows:y = constant + PS + animal ID(1)
where y indicates the agreement (1 = agreement or 0 = no agreement), PS = observed panting scores (0, 1, or 2), and animal ID = individual cattle effect.

The performance of ear tag sensors in reporting panting was determined qualitatively by calculating sensitivity, specificity, and positive predictive values for both raw and fill data as per the following formula:(2)$$Sensitivity=\frac{Count\;of\;True\;Positive\;panting}{Count\;of\left(True\;Positive\;+False\;Negative\right)\;panting}\times 100.$$
(3)$$Specificity=\frac{Count\;of\;True\;No\;panting}{Count\;of\left(True\;Negative+False\;Positive\right)\;panting}\times 100,\;and$$
(4)$$Positive\;predictive\;value=\frac{Count\;of\;True\;Positive\;panting}{Count\;of\left(True\;Positive+False\;Positive\right)\;panting}\times 100.$$

Sensor performance was also assessed by a regression model comparing scatter plots of observed values against sensor-derived values (Observed versus Sensor; [49]). Then, the association between observed and sensor-derived panting duration was evaluated by Pearson’s correlation coefficient (r) and a coefficient of determination (R^2^), which represent the proportion of total variance in the observed data explained by the sensors’ panting report [50]. Lin’s concordance correlation coefficient (CCC) [51] was determined to measure the precision of sensor-derived panting simultaneously with an ideal value fit of 1. Bias correction factor (C_b_) was determined to measure the accuracy of the sensors’ panting duration report that indicated how far the regression line deviated from the concordance line (Y = X). Confidence intervals were determined for the parameters of quantitative association between duration of observed panting and sensor-derived panting. Correlation coefficients (r and CCC) and C_b_ were compared as per the classifications described by Bikker et al. [40] and Pereira et al. [41].

#### 2.3.2. Experiment 2

Sensor-derived minute-level panting data of individual cattle were organised on an hourly basis per day. The average hourly panting minutes per animal was considered as the outcome variable explained by animal factors (genotype, coat colour, body weight, sex, and docility) and climatic factors (temperature, relative humidity, solar radiation, wind speed, and rainfall). To estimate the effect of factors determining variability between individual animals in panting, data were analysed using the restricted maximum likelihood method of mixed models (REML—linear mixed model). To represent the climatic effect, THI, HLI, heat event periods, and hour of day were included as factors. The fixed effects of all animal and climatic factors were assessed using the following model:y = µ + F + animal ID + ε(5)
where y indicates the average hourly panting duration (minutes) per animal; µ indicates constant (overall mean); F indicates any of the animal and climatic factors; animal ID includes individual cattle random effect, and ε indicates random residual error.

Interactions among genotype and coat colour; genotype and hour; coat colour and hour; genotype, hour, and period; and coat colour, hour, and period were also determined. Based on the significance of the fixed effect of individual factors, we restricted the final model to genotype, coat colour, and hour of day as predicting factors. As genotype and coat colour are not independent of each other, we used separate models to estimate the effects of genotype and coat colour on panting outcome across hour of day as described below:y = µ + G + H + G.H + animal ID + ε(6)
and
y = µ + C + H + C.H + animal ID + ε(7)
where y indicates the average hourly panting duration (minutes) per animal; µ indicates constant (overall mean); G indicates genotype (2 levels: less than 40% BI and 50% BI); C indicates coat colour (3 levels: dark, tan, and white), H indicate hour of day (24 levels: 1 to 24 h); animal ID includes individual cattle random effect, and ε indicates random residual error.

To represent individual differences within the same genotype and coat colour categories, coefficients of variation (CV) were calculated and the panting data of the top and bottom 10% of panting animals (selected based on average total panting duration/day) were compared (independent t-test). The relationship between the predicted average panting duration/h and hourly thermal index values were assessed fitting regression models, and differences in panting duration across THI and HLI values were determined by multiple comparisons.

## 3. Results

### 3.1. Experiment 1: Observed and Sensor Data Agreement

The percentage of agreement between observed and sensor-derived panting and non-panting states is shown in Table 6. For both raw and fill datasets, good agreement was found for PS0 and PS2; however, the percentage agreement for PS1 was moderate.

Sensitivity, specificity, and positive predictive values are presented in Table 7. Sensor-derived panting states were compared with observed panting and non-panting states for raw and fill datasets. The overall sensitivity of the sensors for reporting panting was 39% and 59% for the raw and fill datasets, respectively. The sensitivity of the sensors was greater at PS2 (45% and 82%, respectively for raw and fill) compared to PS1 (37% and 54%, respectively for raw and fill) for both raw and fill datasets. The specificity (77% and 65%, respectively for raw and fill) with the upper confidence value (83% and 72%, respectively) represents the sensor’s ability to accurately report PS0. The positive predictive value of raw (79%, upper confidence limit 84%) and fill (77%, upper confidence limit 82%) data confirms that approximately 80% of the sensor-recorded panting was true (observed) panting.

Regression analysis results are presented in Figure 5. Pearson’s correlation coefficient (r), Lin’s CCC, and the C_b_ between observed and sensor data for the duration of panting per animal for the raw and fill datasets are shown in Table 8. Observed and sensor-derived panting duration were highly correlated for both the raw (r = 0.89, R^2^ = 0.78) and fill (r = 0.90, R^2^ = 0.81) datasets. The CCC and C_b_ were 0.45 and 0.50 (respectively) for the raw dataset and 0.76 and 0.85 (respectively) for the fill datasets.

### 3.2. Experiment 2: Animal Variability and Comparison with Thermal Indices

There was no effect of sex (*p* = 0.79), body weight (*p* = 0.79), or docility score (*p* = 0.97) on panting duration/h for individual cattle. Differences in average panting duration/h between genotype and coat colour categories are shown in Figure 6 and Figure 7. Panting duration/h was greater (between 15 and 21 min; *p* < 0.001) for the less than 40% BI cattle type compared to 50% BI (between 12 and 17 min) from 0800 to 1700 h. Similarly, coat colour categories differed significantly (*p* < 0.001) in panting duration/h with white coloured individuals having reduced (between 12 and 16 min) panting from 0800 to 1700 h compared to dark (between 15 and 22 min) and tan (between 13 and 18 min) coloured cattle. Dark and tan coloured cattle reached peak panting levels approximately 1 h earlier (between 1100 and 1200 h) than their white coloured counterparts (between 1200 and 1300 h).

The variations in panting duration between individuals within the same genotype (82% and 84% CV, respectively for less than 40% BI and 50% BI) and coat colour categories (81%, 84% and 85% CV for dark, tan, and white coat colours, respectively) were large. However, cattle in the top and bottom 10% for panting duration showed consistency in panting duration across time. The variation in panting duration between individuals within the same genotype and coat categories is represented as average hourly panting minutes across 24 h for the whole herd, and for cattle in the top and bottom 10% for panting duration (Figure 8). The mean hourly panting duration of cattle during the hottest part of the day (1200 to 1500 h) ranged between 10 and 30 min for individuals of less than 40% BI genotype and less than 10 and 26 min for individuals of 50% BI genotype. Average hourly panting duration during the hottest part of the day (1200 to 1500 h) ranged from greater than 10 to 30 min, respectively, for bottom and top panting 10% individuals within the dark coat colour category, whereas the average hourly panting duration for both tan and white coat colour categories ranged from less than 10 to 25 min, respectively for the top and bottom 10% of panting individuals.

The associations between mean thermal index values/h and duration (minute) of mean herd level panting/h are shown in Figure 9 and Table 9. When panting duration/h were compared with thermal indices (THI and HLI), the panting duration of both genotype categories showed a quadratic increase (*p* < 0.001) with the increase in thermal index values. However, cattle belonging to both genotypes showed greater than 5 min of panting/h when the THI and HLI values were below the reference thresholds. Panting duration increased significantly (*p* < 0.001) from the base duration at the THI value of 64 and HLI value of 63 (Figure 9), and panting was consistently 10 min or greater after the THI and HLI range of 65–69 (Table 9), suggesting the need for lower thresholds for both THI and HLI.

## 4. Discussion

Our work highlights the ability to accurately and autonomously reveal the temporal and spatial variability of individual cattle PS through time. In Experiment 1, we identified strong associations between observed and sensor-derived panting duration for both raw and fill data. The fill approach was evaluated due to the fact that the sensor technology automatically prioritises continuously sampled behaviour into a single state per minute according to a variety of proprietary factors, rather than a simple time scale or similar. An animal may perform several behaviours within a single minute, including heavy breathing, which may not necessarily be recorded as heavy breathing during that minute. Agreement and sensitivity in reporting panting were lower for PS1 as compared to PS2. Specificity and positive predictive values indicate that sensors accurately reported PS0, with the probability of reporting false positives being low. Despite the paucity of data from which to draw comparison with our findings, the sensitivity and specificity of the current work aligns with Wolfger et al. [42] who validated ear tags for rumination (49% sensitivity and 96% specificity). Comparatively lower agreement and sensitivity for PS1 is not necessarily due to error in sensor derived panting. Rather, the error may be attributed to the smaller movement associated with PS1 and associated observation error [52,53,54] regarding transitions between panting and non-panting states on a minute-by-minute basis [55]. However, our results clearly show that the agreement and sensitivity of the sensors in determining panting increases at PS2, and as such, the agreement and sensitivity of the sensors will likely further improve with greater panting score. Overall, our results show raw tag panting duration data to be strongly associated with observed panting duration. However, the ‘fill’ data method improved the accuracy and sensitivity of sensors perhaps due to reducing the impact of the effect of transitions between behaviours within a minute. In line with these findings using ear tags, panting as monitored by neck-mounted sensors has been shown to be highly correlated with core body temperature [39]. As this experiment recorded a maximum panting score of 2, there is potential for a stronger association between observed and predicted higher panting scores, which should be determined in further study.

Experiment 2 reveals the distribution in average panting duration/h across time between genotype and coat colour categories. The current findings are consistent with the positive association between the level of *Bos indicus* (BI) [21,23,43,44] and lightness of the coat [6,56,57,58,59] with heat tolerance. Apart from the variation between genotype and coat categories, individuals within the same genotype and coat characteristics may vary significantly in heat stress responses and will require individual attention [18] for strategic heat stress mitigation. Our results also showed high variability in average panting duration/h between individuals within the same genotype and coat colour categories. For example, the bottom 10% of panting individuals of the 50% BI genotype engaged in panting less than 15% of the time (less than10 min/h) during the hottest part of the day (1200 to 1500 h) as compared to around 50% of time (approximately 30 min/h) for the top 10% of panting individuals during the same period. Similarly, white coat coloured cattle had the lowest average panting duration/h (less than 15 min); however, the top 10% of panting individuals within this category showed greater than 25 min of panting/h during the hottest part of the day. As heat abatement measures require a substantial amount of resource [60], our findings suggest that these costs could be minimised by focusing on susceptible animals [61]—for instance, those identified in the top 10%. In addition, the variability between cattle in duration of panting indicates that such sensors could be effective tools for the identification of heat-tolerant cattle to incorporate in future breeding programs for increased heat resilience. Further work focusing on the mechanisms of internal body temperature regulation by low and high panting individuals based on continuous monitoring by temperature loggers and sensors could eliminate the subjective errors of human observations and also the practicality limitations.

Mean panting duration/h increased significantly at a THI value of 64 and HLI value of 63 with no significant difference in base panting duration for THIs between 56 and 63 and HLIs between 50 and 62. The average panting duration/h of 10 min or greater for both genotypes was detected at average THIs and HLIs of greater than 65–69. However, trends of increased panting duration based on our results represent an average herd level panting duration of greater than 5 min/h at lower THI and HLI values compared to recognised thresholds of 68 to 72 [62] for THI and 71 to 77 [25] for HLI, suggesting that a lower threshold between 65 and 69 for both THI and HLI should be used. It is evident that the 50% BI type cattle showed reduced amounts of panting at the same index values as compared with cattle with less than 40% BI type, highlighting the need for individualised management of heat stress for a mixed breed cattle herd. The substantial variability in panting duration that we revealed here in response to heat between and within cattle types reinforces the importance of grouping of animals based on genotype and coat colour and of individualised monitoring.

## 5. Conclusions

Grouping animals based on genotype and coat colour categories and allocating heat stress mitigation resources on a priority basis may substantially increase the productivity of commercial feedlots. However, variability within cattle group is also important to consider. Ear tag sensors can monitor the panting of individual cattle with acceptable accuracy and precision. Our findings suggest that the difficulty of quantifying the individualised impact of heat stress can be overcome by sensors enabling the automation of heat stress monitoring and the mitigation process. Individual variability within similar genotypes and coat categories of cattle as determined by sensors can now be utilised for the isolation of heat-tolerant and susceptible animals for strategic amelioration and for genetic selection to increase heat resilience in future generations. Further work will now focus on using the suite of sensor-derived behaviours to determine cattle heat coping mechanisms.

## Figures and Tables

**Figure 1 animals-10-01518-f001:**
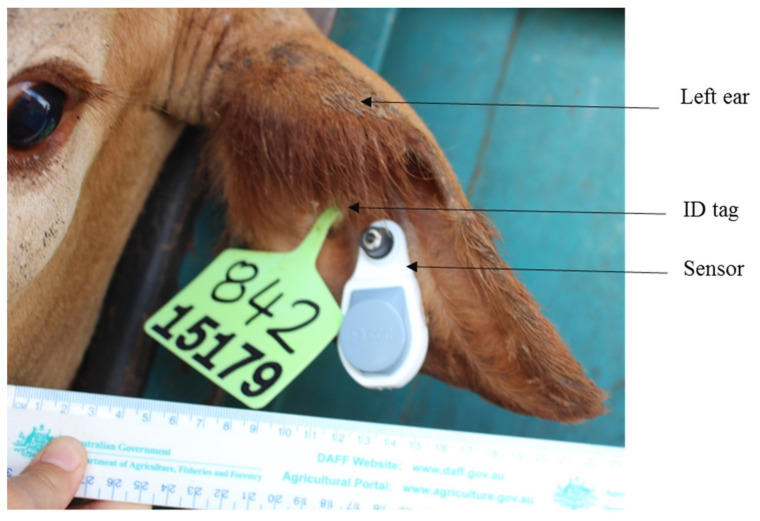
Installation of ear tag sensor.

**Figure 2 animals-10-01518-f002:**
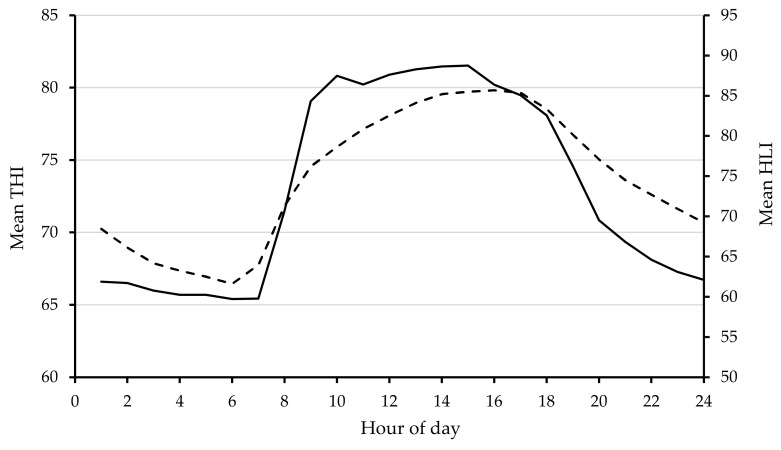
Mean hourly temperature humidity index (THI) (**broken line**) and heat load index (HLI) (**solid line**) values across 24 h for Experiment 1.

**Figure 3 animals-10-01518-f003:**
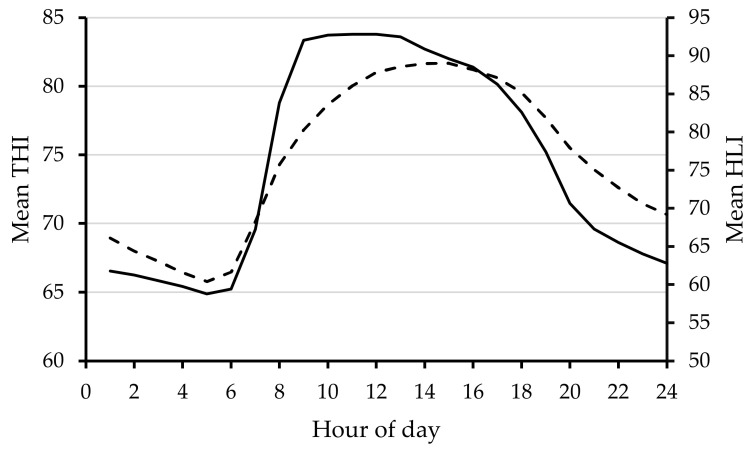
Mean hourly temperature humidity index (THI) (**broken line**) and heat load index (HLI) (**solid line**) values across 24 h for Experiment 2.

**Figure 4 animals-10-01518-f004:**
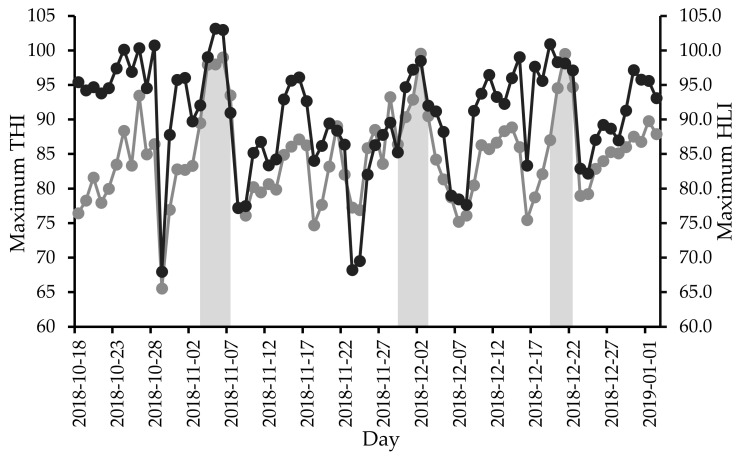
Maximum daily temperature humidity index (THI) (
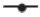
) and heat load index (HLI) (
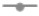
) during Experiment 2. Shaded areas indicate selected heat event periods.

**Figure 5 animals-10-01518-f005:**
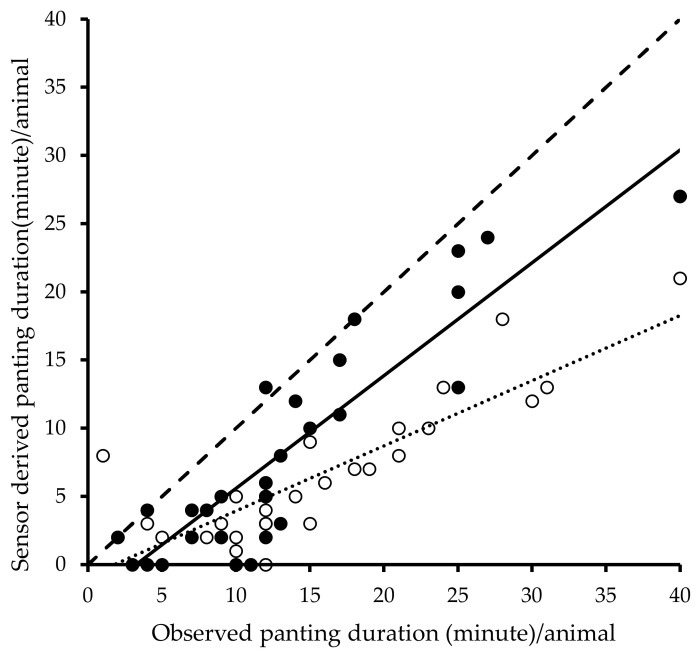
Association between duration (minute) of total observed panting and sensor-recorded panting for raw and fill datasets. The solid line is the equation line for fill data (solid filled circles, y = 0.83x − 2.68, R^2^ = 0.81), the dotted line is the equation line for raw data (circles with no fill, y = 0.48x − 0.84, R^2^ = 0.78), and the dashed line indicates the line of equality (Y = X).

**Figure 6 animals-10-01518-f006:**
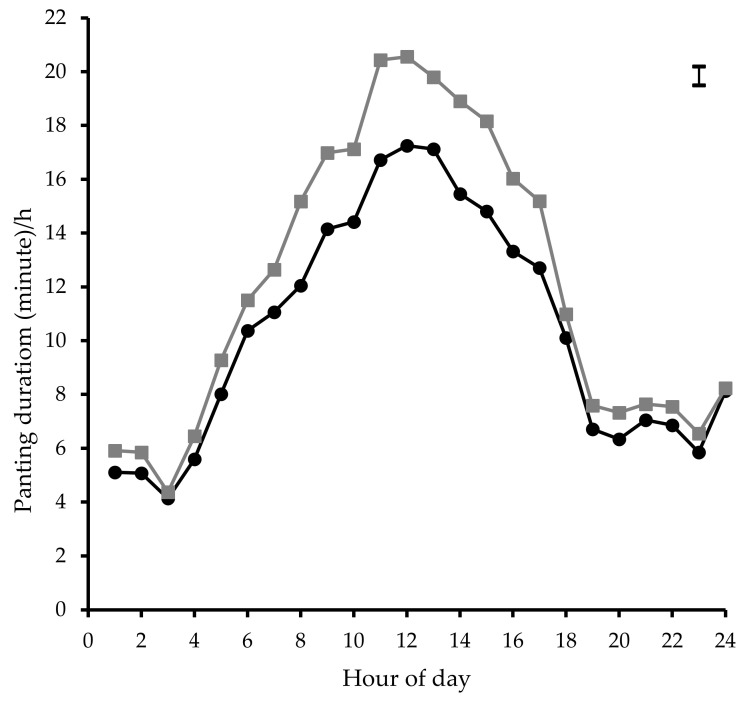
Mean hourly panting duration for less than 40% *Bos indicus* (
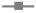
) and 50% *Bos indicus* (
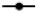
) genotypes within the day. The error bar in the right hand top corner denotes average SED.

**Figure 7 animals-10-01518-f007:**
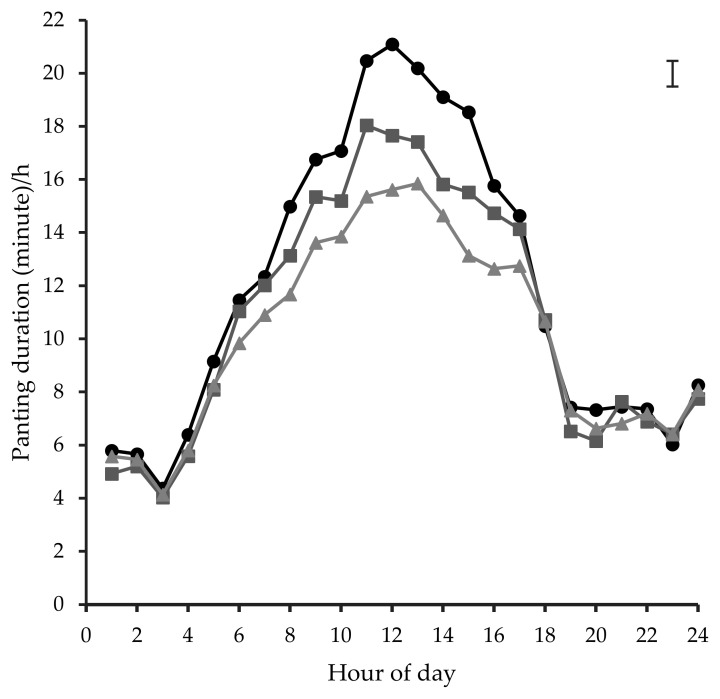
Mean hourly panting duration for cattle of dark (
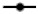
), tan (
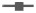
), and white (
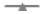
) coat colours within the day. The error bar in the right hand top corner denotes average SED.

**Figure 8 animals-10-01518-f008:**
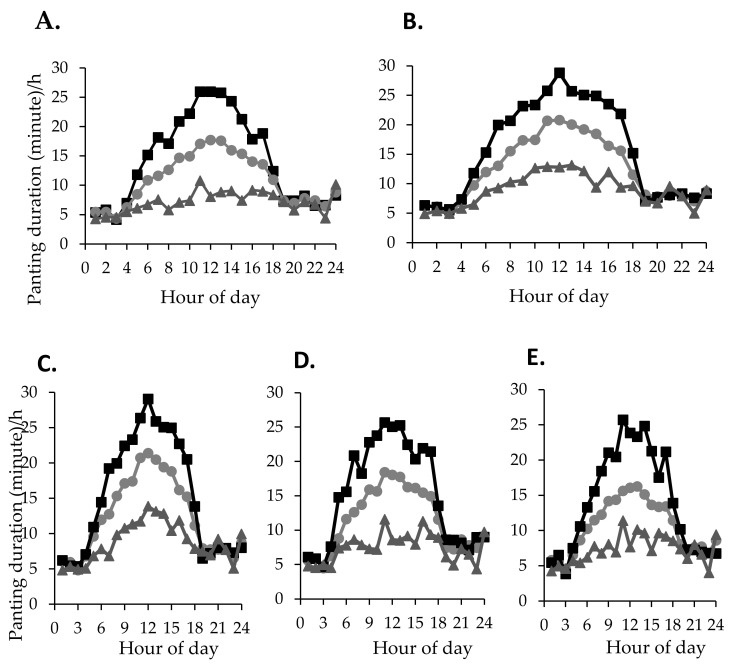
Mean panting duration/h of the top (
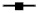
) and bottom (
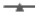
) 10% cattle for panting within the same genotype ((**A**): 50% *Bos indicus* and (**B**): less than 40% *Bos indicus*) and coat colour ((**C**): Dark, (**D**): Tan and (**E**): White) categories in comparison to their herd mean panting duration/h (
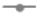
).

**Figure 9 animals-10-01518-f009:**
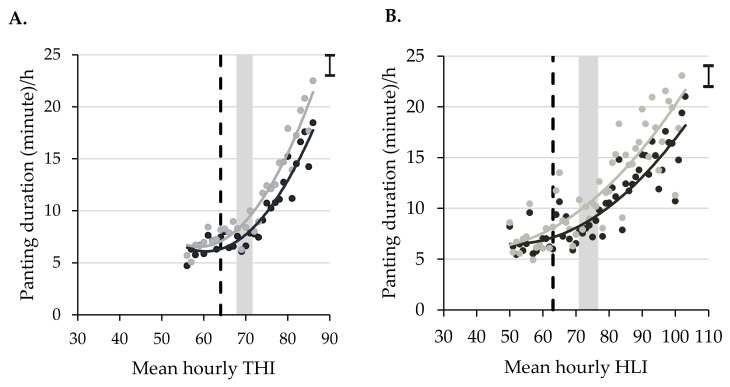
Association between hourly mean thermal index values and mean panting duration (minute)/h of 50% *Bos indicus* (dark grey circles and trend line) and less than 40% *Bos indicus* (light grey circles and trend line) cattle. (**A**) Panting duration versus temperature humidity index (THI) (r = 0.90, y = 0.02x^2^ − 2.2x + 71.84, R^2^ = 0.91 for 50% *Bos indicus*, and r = 0.92, y = 0.02x^2^ − 2.4x + 77.14, R^2^ = 0.93 for less than 40% *Bos indicus*) and (**B**) Panting duration versus heat load index (HLI) (r = 0.86, y = 0.004x^2^ − 0.33x + 12.67, R^2^ = 0.77 for 50% *Bos indicus*, and r = 0.86, y = 0.004x^2^ − 0.43x + 17.40, R^2^ = 0.80 for less than 40% *Bos indicus*). The reference threshold THI and HLI range for heat stress is indicated by shaded zones. The vertical broken lines indicate the point of THI and HLI at which the mean panting duration increased significantly from the base. The error bars in the right hand top corners indicate average SED value for panting duration across THI (**A**) and HLI points (**B**).

**Table 1 animals-10-01518-t001:** Feed composition for the rations offered to cattle in Experiment 1.

Item	Ration 1	Ration 2
Ingredients (%)		
Sorghum hay	27.0	3.0
Silage	10.0	17.0
Biscuit meal	13.0	17.0
Canola meal	5.5	4.5
Barley	39.5	50.5
Oil	-	3.0
Liquid finisher supplement *	5.0	5.0
Composition (%)		
Dry matter (DM)	76.7	71.1
Crude protein (CP)	14.0	13.5
Neutral detergent fibre (NDF)	32.2	23.3
Acid detergent fibre (ADF)	21.4	17.2
Metabolisable energy (ME, MJ/kg DM)	10.9	12.8

* Composed of molasses, urea, limestone, ammonium sulphate, xanthan gum, monensin, flavophospholipol, magnesium, trace minerals and vitamins; containing 70.5% DM, 65.7% CP, 5% calcium, 1% magnesium, 52 IU/kg vitamin A, 215 IU/kg vitamin E, 468 ppm monensin, 39 ppm flavophospholipol and 674 ppm zinc on a DM basis.

**Table 2 animals-10-01518-t002:** Average daily temperature and relative humidity with minimum and maximum values and SD for Experiment 1.

Parameter	Daily Temperature °C			Daily Relative Humidity %		
	Minimum	Mean	Maximum	Minimum	Mean	Maximum
Mean	19	26	33	33	60	81
SD	2.1	1.4	2.7	5.1	3.0	11.0

**Table 3 animals-10-01518-t003:** The respiratory dynamic and panting score (PS) of cattle as modified from Gaughan et al. [25], Mader et al. [26], and Brown-Brandl et al. [17].

PS	Description *
0	Normal breathing, no forward–backward heaving. Respiration rate less than 60 breaths/min.
1	Forward–backward heaving, mouth closed, no drool or foam, easy to see chest movement. Respiration rate between 60 and 100 breaths/min.
2	Forward–backward heaving, mouth closed, but drool or foam present. Respiration rate between 100 and 120 breaths/min.
3	Forward–backward heaving, mouth open or intermittent mouth open, excessive drooling, tongue not extended, neck extended, and head held up. Respiration rate between 120 and 160 breaths/min.
4	Forward–backward heaving, open mouth with tongue protruding either occasionally or for prolonged periods, excessive drooling, neck extended, head held up or down. Respiration rate greater than 160 breaths/min and may be variable due to phase shift in respiration.

* Panting scores were assigned based on visual observation of respiratory dynamic and behaviour, not on the estimation of respiration rate.

**Table 4 animals-10-01518-t004:** Feed composition for the rations offered to cattle in Experiment 2.

Items	Ration 1	Ration 2	Ration 3	Ration 4
Ingredients (%)				
Oaten hay	3.5	3.0	2.0	-
Cotton hulls	12.5	10.0	8.5	3.0
Barley silage	29.0	25.0	15.0	-
Sorghum silage	-	-	-	12.0
Whole cotton seed	13.0	13.0	13.0	13.0
Wheat grain	36.6	43.0	54.7	-
Barley grain	-	-	-	63.8
Oil	1.0	1.5	2.0	3.2
Liquid finisher supplement *	4.4	4.5	4.8	5.0
Composition (%)				
Dry matter (DM)	68.1	69.7	73.6	74.2
Crude protein (CP)	13.1	13.3	13.4	14.1
Neutral detergent fibre (NDF)	38.3	34.1	29.0	27.4
Acid detergent fibre (ADF)	27.8	24.7	20.8	19.6
Metabolisable energy (ME, MJ/kg DM)	11.3	11.8	12.6	14.1

* Same as indicated in the footnote of Table 1.

**Table 5 animals-10-01518-t005:** Mean daily temperature and relative humidity with minimum and maximum values and SD for Experiment 2.

Parameter	Daily Temperature °C			Daily Relative Humidity %		
	Minimum	Mean	Maximum	Minimum	Mean	Maximum
Mean	19	27	35	30	60	85
SD	3.4	2.6	3.1	9.5	7.4	3.5

**Table 6 animals-10-01518-t006:** Percentage of agreement (minute by minute) between observed panting score (PS) and sensor-recorded panting states for raw and fill datasets.

PS	Agreement (%)	
	Raw Data	Fill Data
0	75	65
1	45	54
2	68	83

**Table 7 animals-10-01518-t007:** Sensitivity, specificity, and positive predictive value (%) of sensor-derived panting states (minute by minute) compared to observed panting score (PS) for raw and fill datasets.

Parameter	Raw Data	Fill Data
Sensitivity (%)		
Overall	39 (35–43) *	59 (54–63)
PS = 1	37 (33–42)	54 (49–60)
PS = 2	45 (35–55)	82 (72–92)
Specificity (%)	77 (72–83)	65 (58–72)
Positive predictive value (%)	79 (73–84)	77 (72–82)

* Values within parentheses indicate 95% confidence interval.

**Table 8 animals-10-01518-t008:** Correlations between duration (minute) of total observed panting and sensor-recorded panting.

Parameter	Raw Data	Fill Data
Pearson’s correlation coefficient, r	0.89 (0.86–0.92) *	0.90 (0.87–0.93)
Lin’s concordance correlation coefficient, CCC	0.45 (0.29–0.58)	0.76 (0.60–0.86)
Bias correction factor, C_b_	0.50 (0.34–0.63)	0.85 (0.69–0.93)

* Values within parentheses indicate 95% confidence interval.

**Table 9 animals-10-01518-t009:** Mean herd level panting duration (minute) for 50% *Bos indicus* and less than 40% *Bos indicus* cattle across mean hourly THI and HLI classes.

Mean Panting Duration	THI/HLI Class											SED	*p* Value
	50–54	55–59	60–64	65–69 *	70–74	75–79	80–84	85–89	90–94	95–99	100–104		
THI: 50% BI **	-	5 ^a^	7 ^b^	9 ^c^	10 ^c^	13 ^d^	16 ^e^	18 ^f^	-	-	-	0.43	<0.001
Less than 40% BI	-	5 ^a^	8 ^b^	11 ^c^	11 ^c^	15 ^d^	19 ^e^	21 ^f^	-	-	-	0.44	<0.001
HLI: 50% BI	6 ^a^	8 ^b^	8 ^b^	9 ^c^	10 ^d^	11 ^e^	12 ^f^	15 ^g^	17 ^i^	16 ^h^	15 ^g^	0.40	<0.001
Less than 40% BI	6 ^a^	9 ^b^	9 ^b^	9 ^b^	10 ^c^	12 ^d^	15 ^e^	17 ^f^	20 ^i^	19 ^h^	18 ^g^	0.40	<0.001

^a–i^ indicate panting duration within the same row with a common letter that did not differ significantly. * indicate the THI/HLI range after which panting durations were consistently ≥10 min for both genotypes. ** BI indicates *Bos indicus*.

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
