# Peer review of "Automated Monitoring of Panting for Feedlot Cattle: Sensor System Accuracy and Individual Variability"

_animals, 2020, doi:10.3390/ani10091518_

Round 1

Reviewer 1 Report

Review animals 855082  Ear tag panting sensor

I am concerned about the presentation and conclusions of the results, in that authors seem to be omitting some useful results in their goal to promote the product.   For example, the Abstract is largely dedicated to their comparison and detailed analysis of raw vs fill data.  Which is of limited value compared to the absence of anything supporting their conclusion sentence for the abstract, which is reproduced below in quotes.  When I look for data supporting that conclusion, what jumps out from Fig 5,6,7 and Table 9 is that breed and coat colour make a tremendous difference, as is nicely shown in this study – so this study has value!!  Yet these authors only emphasize that using their methods, there is some variation between individuals within their classification of breed and colour.  Fine, they can mention that there is some variation which can make their product useful for certain applications, but what about the very obvious results that breed and colour are also important?    A logical conclusion that this paper needs to also make is that grouping cattle by breed and coat colour will allow management to treat animals for heat stress on a group basis(put the most heat stress susceptible cattle in the pens with the sprinklers or best shade), which is much easier and realistic in feedlots.  With careful re-wording and editing (being more concise), I think these authors can do both.

"However, there was high variability (CV > 80%) in the duration of panting between individuals within same breed and same coat colour revealing the potential for strategic management at an individual level, and with the collation of data, genetic selection for heat resilience."

General suggestions:

Jumping from Exp 1 to Exp 2 and back again through the paper is confusing.  It would be better to present Exp 1 in its entirety (methods, results, discussion), then Exp 2.  The distinction of Exp 1 and 2 is not used in the results and discussion making it hard for readers.

Could panting duration be biased when comparing “real” aka, Observed, with Sensor, because observers stopped at 15 min per subject or when an animal moved out of sight?  Hence, some durations could be suspect (cut off), which needs to be addressed in the Discussion section.  It would also be appropriate to refer to the Observed as Observed, and not “real”.

21           should describe what is meant by “reasonable”.

31           Minute states?? Vague.

38           fill data?  Why?  2 min?

49           is that fill data?

98           mixed?  Do you mean mixed breeds?

127         “inducted” is not commonly used term.  I understand, but why not just relate it to day 1, 2, etc. as you later did?

129         I tried website, but it seems to have changed and it may need to be updated. 

131         explain or summarize manufacturers guidelines so readers have an idea what is involved.

131         “optimized”?  please explain.

  133       not clear what “these sensors” refers to?  ID tag?  How “optimized”?

139         another place where I suggest replacing “inducted”.

167         briefly described THI, ref 41, so should also describe HLI, ref 18.

167-169                and other places in paper.  As a matter of style, it is much nicer to introduce tables and figures in the text in a less wordy way than reading the table of figure title, followed by “are shown in”.

186         why use the term “behaviours” here and elsewhere when you only worked with “panting” or “other”?  Readers look for the behaviours, but none are reported.

190         need some discussion about inter and intra observer reliability.  This is a requirement for most ethological studies. I would be nice if the observers looked at the same subject several times and their agreement of their subjective scores was determined.  Most good ethology/behavior books have a discussion of this topic. 

201         what is meant by “according”?  Not clear what sampling was based on. 

213         confusing.  The first thing every PS is defined by in the table is respiration rate, yet authors claim that PS was “not on the estimation of respiration rate”.

221 – 222             the authors really just collected panting and not panting – which is fine, but need to revise paper to make this clear.  The authors may have initially intended to do more with “behavior” categories and then lumped them together, which happens a lot. 

234         what is collated?  Just drop the use of such unnecessary terms and make the paper more succinct. 

237         Figure 3 is helpful, help readers understand.

268         because filled data seemed to be more useful, why not use filled data in Exp. 2?  An explanation of why “filled” was not used would help.

268         suggest you just say that experimental unit was mean hourly observations of panting, or something similar??  Table 7 title says “minute by minute”, so confusing.  304 uses the term “minute states”, which is not clear, so standardizing by using clear terms will help.

444         these authors are concerned about “subjective errors of human observations”, but base the results on human observations with little to support the reliability of their observations.  This apparent contradiction needs to be addressed.

456 and 462        It is hard to treat an individual within large groups.  Why not stress the importance of grouping by genotype & coat colour and treat by group?  Of course, the product will help identify exceptional individuals.

Figures 5,6 & 7   Plotting “sensor” and “true” would be useful in these figures, but not sure if these are Exp 1 or 2 results.  Showing these relationships graphically would be most useful.  This is also an example of why it would be nice if the paper was organized by experiment.

Reviewer 2 Report

General Comments

The concept of this study is great, I enjoyed reading it. I have some expansive comments for the introduction and materials and methods. Specifically, I would like to see less ambiguity in the introduction and have offered some specific comments to provide direction with this. Similarly, there are some key concepts relating to heat stress discussed throughout the introduction that are not novel, yet are poorly cited. I would like to see this corrected. I have some suggestions for improving data presentation particularly in the figures. I have no concerns with the discussion, I think you’ve done a great job of interpreting and explaining your results. Just a note the template used to prepare the manuscript is the 2018 version, you might want to update to the 2020 version. Overall you’ve presented a really nice study here, well done

Summary

Line/s 14. Please consider revising ‘physiological and behavioural responses’

Line/s 15. On a commercial scale is impractical

Introduction.

Line/s 65-67. Please consider revising ‘physiological and behavioural responses’ to just “animal responses’ I understand why you want to use ‘physiological and behavioural’ but it is probably more appropriate to not define them at all. We often classify PS into behaviour I suspect due to the nature of data collection but in reality it is actually a physiological response, hence my suggestion here. Also what do you mean by high temperature’s here? Can you quantify this some more? I don’t think that you need to have ‘an associated environmental and animal factors’ here as that has been defined previously

Line/s 69. Can you provide a citation for the thermal exchange pathways, I don’t believe that this is a novel concept thus needs a citation

Line/s 70. At what point do non-evaporative mechanisms become less effective?  Can you quantify this some more? Similar the previous comment this is not a novel concept and I would like to see this cited.

Line/s 76-77. Difficulty in commercial situations? Please see Gaughan JB, Mader TL, Holt SM, Sullivan ML, Hahn GL(2010b) Assessing the heat tolerance of 17 beef cattle genotypes. International Journal of Biometeorology 54(6), 617–627. doi:10.1007/s00484-009-0233-4

Line/s 78-81. The climate indices weren’t really developed because of the lack of accuracy in respiratory dynamics ect, they have been developed due to the inadequacy of relying on a singular climatic variable specifically ambient temperature.

Line/s 84. There are numerous authors that have suggested this.

Line/s 91. Please change were to are. And this sentence requires a formal citation

Line/s 92 – 95. These measures have always been incorporated into the HLI

Line/s 96-98. No the HLI doesn’t but the AHL does, which was developed and is used in conjunction with the HLI in commercial situations and in some research

Line/s 99. The assessment of what?  

Line/s 109-111. There is newer manuscript out from the Gaughan Lab that has started working on this. Please see Lees, A. M., J. C. Lees, V. Sejian, M. L. Sullivan, and J. B. Gaughan. 2020. Influence of shade on panting score and behavioural responses of Bos taurus and Bos indicus feedlot cattle to heat load. Animal Production Science 60(2):305-315. doi: 10.1071/AN19013

Materials and methods.

Line/s 120-123. I think that the purpose of the experiments could be separated into two sentences.

Line/s 127. Please present as ‘with a liveweight of 378 ± 28.7 kg. And I would ask why SD and not SE? – same thing for Exp 2

Line/s 125 -134. Perhaps a figure (photo) show tag placement would be useful, OA online journal it won’t impact APC for this manuscript  

Line/s 137-138. Please provide more information here, going off this I wouldn’t be able to go out and categorise cattle in a similar fashion. Again perhaps some form of diagram would be useful here

Line/s 139-140. What is this docility score – please provide more detail

Line/s 141- 144. This information would be better presented on line 139 after the description of BI content classifications

Line/s 166. Out of interest sake why did you choose this THI equation? There is one that was first defined by Davis et al that is predominantly used in feedlot studies. Please see Davis, M. S., T. L. Mader, S. M. Holt, and A. M. Parkhurst. 2003. Strategies to reduce feedlot cattle heat stress: Effects on tympanic temperature. Journal of Animal Science 81(3):649-661. doi: 10.2527/2003.813649x. Note – I’m not asking you to rework calculations to this equation as that would result in a total reanalysis, I’m just interested to know why you went with the NRC equation when most don’t.

Line/s 191-193. At what interval were these data collected?

Line/s 197. These were steers were they not?  

Line/s 239. What software was used to conduct the statistical analysis? Lines 298-300, this information should really come at the start of the statistical analysis section

Line/s 271. Please remove ect.

Line/s 278. Please refrain from using cow – were talking about feedlot cattle

Discussion

Line/s 412-414. It could also be associated with smaller movement associated with respiration at a PS 1

Data Presentation

I think tables 3+4 could be combined.

Figures 1 + 3. I think it would be best to shift one index to the secondary axis, then modifiy the data ranges to suit – in their current for its difficult to completely evaluate what is happening as all the data is bunched together. Similar thing for Figure 3.

Reference/s

Citation 15. Is MLA I think that there is plenty of information available in the formal literature that this is an unnecessary citation, so I would suggest removing this as you’ve got the key citations here for PS (18-20).

Round 2

Reviewer 2 Report

General Comments

I thank the authors for considering my previous comments. I have a few additional queries. Also just a note to proof read as there are some typos throughout.

Line/s 4. You might want to change Amanda k. Doughty to Amanda K. Doughty

Line/s 15. PS aren’t necessarily evoked because of extremes – generally ‘hot’ days cattle will pant particularly in the location

Line/s 102-104. Actually this is the whole point of the AHL – it is a function of intensity x duration and the thresholds for the HLI are reflective of different genotypes.

Line/s 119. Where does one third come from?

Line/s 145-146. I don’t think this is necessary, this has been covered in your introduction already and as such its not pertinent to your M&M

Line/s 161. 2.? Please check line 177 as well

Line/s 169. 0800 rather than 8000

Line/s 231. I don’t feel “Indicus” is appropriate. Who’s source factsheet?

Line/s 234-236 – what does ‘dark’ mean red? Black?

Line/s 264 – what is morning, noon and afternoon. You use 24 hour time earlier please use here as well. Also visual observation of what?
